# Role of mHealth applications for emergency medical system activation in reducing mortality in low-income and middle-income countries: a systematic review protocol

Dylan Griswold [1,2] Andres M Rubiano [3,4]

¹Clinical Neurosciences, Cambridge University, Cambridge, UK
²Stanford University School of Medicine, Stanford, California, USA
³Center for Research and Training in Disaster Medicine, Humanitarian Aid and Global Health, Università degli Studi del Piemonte Orientale Amedeo Avogadro Scuola di Medicina, Novara, Italy
⁴Clinical Research, Meditech Foundation, Cali, Colombia

**Correspondence to**
Professor Andres M Rubiano;
andresrubiano@aol.com

## ABSTRACT

**Introduction** Prehospital care is an essential component in reducing mortality for patients presenting with emergency medical conditions. Prehospital systems tend to be underdeveloped or non-existent in these areas, with less than 1% of low-income and middle-income country (LMIC) populations served by an organised prehospital system. Mobile health apps for activation of Emergency Medical System (EMS) have been shown to decrease mortality, but there has yet to be a systematic review and meta-analysis performed to clarify the role that these apps play in reducing mortality in LMICs. The objective of this review is to evaluate the effectiveness of mobile health apps for EMS activation versus traditional EMS dispatches in survival and transport time in patients with emergency medical conditions.

**Methods and analysis** The proposed systematic review of randomised controlled trials (RCTs) and non-randomised controlled trials (NRCTs) will be conducted in accordance with the Joanna Briggs Institute methodology for systematic reviews of effectiveness evidence. MEDLINE, CINAHL, Web of Science, Cochrane Library, EMBASE and EBSCO will be searched from January 2005 to March 2021. The search results will be presented according to the Preferred Reporting Items for Systematic Reviews and Meta-Analyses flow diagram. Primary outcomes will include mortality and transport time. Critical appraisal will be assessed using the JBI SUMARI Risk-of-Bias Tool for RCTs, and Risk-of-Bias In Non-Randomised Studies tool for NRCTs. A narrative synthesis will be conducted for all included studies. If sufficient data are available, a meta-analysis will be conducted. I² statistics will be used to assess heterogeneity and identify their potential sources.

**Ethics and dissemination** No ethical approval will be required, as this review is based on already published data and does not involve interaction with human subjects. The plan for dissemination, however, is to publish the findings of the review in a peer-reviewed journal and present findings at high-level international conferences that engage the most pertinent stakeholders. Any amendments to this protocol will be documented in the final review.

**PROSPERO registration number** CRD42021243041.

## Strengths and limitations of this study

⇒ To the best of our knowledge this protocol provides a detailed description of the first systematic review on the effectiveness of mobile health apps for Emergency Medical System activations in reducing mortality in low-income and middle-income countries.

⇒ The protocol adheres to the Preferred Reporting Items for Systematic Reviews and Meta-Analyses Protocols guidelines for reporting a systematic review protocol.

⇒ Given the niche nature of the topic it is possible there may not be enough data to perform a meta-analysis.

## INTRODUCTION

Prehospital care is an essential component in reducing mortality for patients presenting with emergency medical conditions. Prehospital systems tend to be underdeveloped or non-existent in these areas, with less than 1% of low-income and middle-income country (LMIC) populations served by an organised prehospital system.[1] Mobile health (mHealth) apps for Emergency Medical Systems (EMSs) activation have been shown to decrease mortality, but EMS development in many LMICs is in its infancy and remains underdeveloped compared with the already existing need for better prehospital care and transport.[2–5] A literature review evaluating the status of EMS in 16 LMICs found that most of the included countries lacked an organised prehospital system and were greatly hindered by financial constraints.[6] Another constraint was found to be a lack of the importance of an EMS among locals. The review found that countries with well-developed prehospital care like the USA and the UK had the most influence in the development of well-functioning EMS through

the implementation of training programmes, support of local EMS leadership, advocacy and awareness and funding.

mHealth services hold promises in helping bridge the gap between the organisation of a functional EMS and a well-functioning prehospital system. There is a wide range of cheap, easily accessible tools which can be used at the point of dispatch or care delivery. With a rapid increase in cellular services, health workers have new opportunities to reach and treat patients who they could not before. There has been an increase in the number of mHealth projects currently active in LMICs, but these tend to be pilot studies that have not been scaled up.[7] The good news is that with COVID-19 came a surge in implementation of mHealth strategies to improve pandemic management and response. Countries in which these strategies were used for planning, surveillance, testing, contact tracing, quarantine and healthcare were more successful.[8] Countries that can leverage this existing network to improve their EMS will likely find a noticeable improvement in prehospital care soon.

While it is unlikely there is a large body of studies examining the effect of mHealth apps for EMS activation regarding mortality and transport time in LMIC populations, it is important to gain a better understanding of the studies that have been published and for which there is data worth appraising. A systematic review and meta-analysis to clarify the role mHealth apps for EMS activation play in reducing mortality in LMICs has not yet been written. Our objective is to carry this out to provide data that will help improve mHealth apps for EMS dispatch in LMICs.

## METHODS

A protocol of this systematic review following the Preferred Reporting Items for Systematic Reviews and Meta-Analyses (PRISMA) statement was registered in the International Prospective Registry of Systematic Reviews. Any changes to the protocol will be amended in PROSPERO and reported in the final review. This systematic review was conducted following the JBI methodology for systematic reviews and meta-analyses.[9 10] This protocol adheres to the Preferred Reporting Items for Systematic Reviews and Meta-Analyses protocols (PRISMA-P) 2015.[10 11]

### Patient and public involvement

Patients and the public were not involved in the design of this systemic review protocol.

### Study design

A systematic review of peer review literature following the PRISMA approach by Moher *et al* is planned for this review.[10 11] Figure 1 summarises the planned stages of the review as described in this protocol.

### Data source and search strategy

The search strategy will aim to locate both published and unpublished studies, including the grey literature. An initial limited search of MEDLINE (via PubMed) and CINAHL was undertaken to identify articles on the topic. The text words contained in the titles and abstracts of relevant articles, and the index terms used to describe the articles will be used to develop a full search strategy for Web of Science, Cochrane Library, EMBASE and CINAHL (EBSCO) will be searched from January 2005 to March 2021 to reflect contemporary practice (online supplemental 1). There will be no restriction regarding the language. The search strategy, including all identified keywords and index terms, will be adapted for each included database and/or information source. The reference list of all included sources of evidence will be screened for additional studies. EndNote V.20 will be used to store, organise and manage all references.

The existence of controlled descriptors (MeSH terms and CINAHL headings) and their key words was verified in each database. The search terms were combined using the Boolean operators 'AND' and 'OR.'[12]

A search strategy combining MeSH terms and free-text words such as—(mobile OR telemedicine* OR ehealth OR telehealth OR smartphone* OR 'smart phone*') AND ('emergency medical service*' OR 'emergency medical system*' OR 'emergency health service*' OR EMS) AND Cochrane's LMIC search filter.[13]

### Selection of studies

Following the search, all identified citations will be collated and uploaded into EndNote V.20 (Clarivate Analytics, Pennsylvania, USA). The citations will then be imported into JBI SUMARI for the review process. Two independent reviewers will examine titles and abstracts for eligibility. The full text of selected studies will be retrieved and assessed. Full-text studies that do not meet the inclusion criteria will be excluded, and a list of such excluded studies will be provided. Disagreements between the reviewers during title and abstract screening or full-text screening will be resolved by consensus, or with a third reviewer. The results of the search will be reported in full in the final report and presented in a PRISMA flow diagram.

### Eligibility criteria
Inclusion criteria

### Participants

This review will include patients of any age with emergency medical conditions including but not limited to stroke, myocardial infarction, trauma, and respiratory failure.

### Intervention(s)

mHealth EMS dispatch, alerts, triage systems and applications.

### Comparator(s)

Traditional EMS dispatch, alerts, and triage systems.

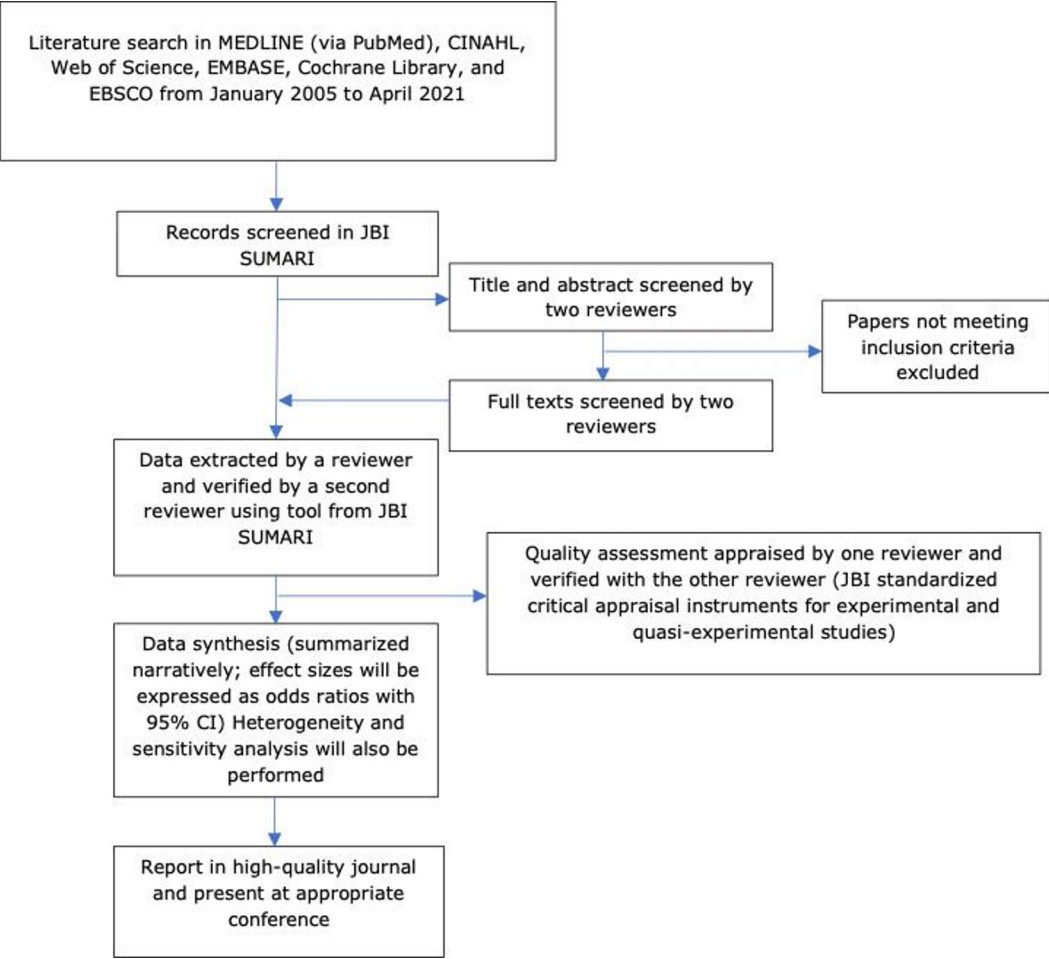

**Figure 1** Summary of search strategy process.

## Outcomes

This review will consider studies that include the following outcomes: mortality and transport time expressed as incidence differences, risk ratios or ORs. All outcomes will be summarised narratively.

## Types of studies

This review will consider both experimental and quasi-experimental study designs including randomised controlled trials, non-randomised controlled trials, before and after studies and interrupted time series studies. In addition, analytical observational studies including prospective and retrospective cohort studies, case–control studies and analytical cross-sectional studies will be considered for inclusion. This review will also consider descriptive observational study designs including case series, individual case reports and descriptive cross-sectional studies for inclusion. We will include preprint studies identified in our search, but no ongoing studies will be considered.

## Assessment of methodological quality

Eligible studies will be critically appraised by two independent reviewers at the study level using standardised critical appraisal instruments from the Joanna Briggs Institute for experimental and quasi-experimental studies. Authors of papers will be contacted to request missing or additional

data for clarification, where required. Any disagreements that arise will be resolved through discussion, or with a third reviewer. The results of critical appraisal will be reported in narrative form and in a table.

All studies, regardless of the results of their methodological quality, will undergo data extraction and synthesis (where possible).

## Data extraction

Data will be extracted from studies included in the review by two independent reviewers using the standardised data extraction tool. The data extracted will include specific details about the populations, study methods, interventions and outcomes of significance to the review objective. Any disagreements that arise between the reviewers will be resolved through discussion, or with a third reviewer. Authors of papers will be contacted to request missing or additional data, where required.

## Data synthesis

Studies will, where possible be pooled in statistical meta-analysis using JBI SUMARI. Effect sizes will be expressed as either odds ratios (for dichotomous data) and weighted (or standardised) final postintervention mean differences (for continuous data) and their 95% CIs will be calculated for analysis. Heterogeneity will be assessed statistically using the

standard $\chi^2$ and $I^2$ tests. Statistical analyses will be determined by the final heterogeneity in the included studies.[13] Sensitivity analyses will also be conducted. Where statistical pooling is not possible, the findings will be presented in narrative form including tables and figures to aid in data presentation where appropriate. A funnel plot will be generated in Microsoft Excel (Seattle, Washington, USA) to use to assess publication bias if there are 10 or more studies included in a meta-analysis. Statistical tests for funnel plot asymmetry (Egger test, Begg test, Harbord test) will be performed, where appropriate.

### Assessing certainty in the findings

The Grading of Recommendations, Assessment, Development and Evaluation approach for grading the certainty of evidence will be followed.[14] The Summary of Findings (SoF) will present the following information, where appropriate: absolute risks for the treatment and control, estimates of relative risk and quality of the evidence based on the risk of bias, directness, heterogeneity, precision and risk of publication bias. The outcomes reported in the SoF will be mortality and transport time.

### Ethics and dissemination

No ethical approval will be required, as this review is based on already published data and does not involve interaction with human subjects. The plan for dissemination, however, is to publish the findings of the review in a peer-reviewed journal and present findings at high-level international conferences that engage the most pertinent stakeholders. Any amendments to this protocol will be documented in the final review.

### DISCUSSION

This protocol has been rigorously developed and designed specifically to assess the effects of mHealth apps for EMS activation on mortality and transport time in LMICs. Given the likely limited evidence associated with the primary objective, findings from the review will be critical for researchers, policymakers, government and non-governmental organisations for planning, developing and improving mHealth apps for EMS activation in LMICs. If protocol modifications are required, the authors will include the detailed description of any changes along with a justification during the publication of the review.

The main barriers for implementing mHealth apps for EMS activation include regulatory bodies, technological expertise and user interaction. Usability of the application, signal loss, data volume utilisation, need to enter passwords and the availability of automated or in-app context-relevant clinical advice are important considerations for the three groups involved: experts, front-line healthcare workers and patients. This review will serve as an important role as a repository of available evidence for the purpose of addressing these barriers and setting effective policy and guideline recommendations.

**Acknowledgements** This article is the result of a study conducted in the framework of the International PhD in Global Health, Humanitarian Aid and Disaster Medicine jointly organised by Università del Piemonte Orientale (UPO) and Vrije Universiteit Brussel (VUB). The authors would like to thank Stanford University research librarian, Lily Ren for her invaluable contribution in helping to develop a relevant and robust search strategy.

**Contributors** AMR conceived the review. DG and AMR refined the review design. Both authors were involved in subsequent draft manuscript reviews and updates and approved the final version of this protocol.

**Funding** DG was supported by the Gates Cambridge (Grant #: OPP1144) and Dr Rubiano was supported by the CRIMEDIM centre at Università del Piemonte Orientale (UPO) (Grant #: N/A).

**Competing interests** None declared.

**Patient consent for publication** Not applicable.

**Provenance and peer review** Not commissioned; externally peer reviewed.

**ORCID iDs**
Dylan Griswold http://orcid.org/0000-0003-0291-8360
Andres M Rubiano http://orcid.org/0000-0001-8931-3254

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
