## [Reviewer comments · BMJ Open]

ARTICLE DETAILS

TITLE (PROVISIONAL)	The role of mHealth applications for Emergency Medical System activation in reducing mortality in low-and middle-income countries: a systematic review protocol
AUTHORS	Griswold, Dylan; RUBIANO, ANDRES M

VERSION 1 – REVIEW

REVIEWER	Ahmed Alanazy University of New England, School of Rural Medicine
REVIEW RETURNED	22-Jun-2021

GENERAL COMMENTS	It Is an interesting Investigation . You have done a good and a comprehensive effort. I have A minor suggestion for the search term. it might add some result for the review . The suggested search terms pre-hospital , out of hospital, Paramedic and EMT.
--

REVIEWER	Elaine Cole Queen Mary University of London, Trauma Sciences
REVIEW RETURNED	28-Jun-2021

GENERAL COMMENTS	It isn't clear as to why trauma has been singled out in the introduction, especially as this SR is focusing on emergency medical conditions (with more named in the methodology section). In relation to trauma, reference 1 is 16 years old. The objective of the systematic review is somewhat lost in the final (long) sentence of the introduction. Suggest rewriting to enhance clarity. Given the likely heterogeneity of the included studies, are you certain that you will use a fixed effect model for the meta-analysis? It may be that you are suspecting a small number of studies in the final review however this isn't clear - not your choice justified.
---

VERSION 1 – AUTHOR RESPONSE

Reviewer: 1

Mr. Ahmed Alanazy , University of New England

Comments to the Author:

It Is an interesting Investigation . You have done a good and a comprehensive effort.

I have A minor suggestion for the search term. it might add some result for the review .

The suggested search terms pre-hospital , out of hospital, Paramedic and EMT.

Thank you, Mr. Ahmed Alanzy, for your comments. As we are already in the full-text review process, we will not change the search strategy at this time.

Reviewer: 2

Dr. Elaine Cole, Queen Mary University of London

Comments to the Author:

It isn't clear as to why trauma has been singled out in the introduction, especially as this SR is focusing on emergency medical conditions (with more named in the methodology section). In relation to trauma, reference 1 is 16 years old.

The objective of the systematic review is somewhat lost in the final (long) sentence of the introduction. Suggest rewriting to enhance clarity.

Given the likely heterogeneity of the included studies, are you certain that you will use a fixed effect model for the meta-analysis? It may be that you are suspecting a small number of studies in the final review however this isn't clear - not your choice justified.

Thank you, Dr. Elaine Cole for your comments. We agree that trauma should not be singled out in the introduction, as it is not singled out in the formal review. We have removed that sentence and the subsequent 15-year old citation that went along with it. We have split the final sentence of the introduction into two sentences. We have amended the method of statistical analyses, noting that the decision will depend on the final heterogeneity of the included studies.